# Effectiveness and Coverage of Treatment for Severe Acute Malnutrition Delivered by Community Health Workers in the Guidimakha Region, Mauritania

**DOI:** 10.3390/children8121132

**Published:** 2021-12-04

**Authors:** Pilar Charle-Cuéllar, Noemí Lopez-Ejeda, Hassane Toukou Souleymane, Diagana Yacouba, Moussa Diagana, Abdias Ogobara Dougnon, Antonio Vargas, André Briend

**Affiliations:** 1Action Against Hunger, 28002 Madrid, Spain; avargas@accioncontraelhambre.org; 2Doctorate Program in Epidemiology and Public Health, Rey Juan Carlos University, 28933 Madrid, Spain; 3Faculty of Biological Sciences, Complutense University of Madrid, 28040 Madrid, Spain; noemilop@ucm.es; 4Action Against Hunger, Nouakchott 1990, Mauritania; hsouleymane@mr.acfspain.org; 5Faculty of Science, Technology and Medicine, Nouakchott University, Nouakchott 880, Mauritania; dyacouba11@gmail.com; 6Maternal and Child Health, Ministry of Health in Mauritania, Nouakchott 115, Mauritania; moussalatou@yahoo.fr; 7Action Against Hunger West and Central Office, Dakar 29621, Senegal; adougnon@wa.acfspain.org; 8Department of Nutrition, Exercise and Sports, Faculty of Science, University of Copenhagen, 1958 Copenhagen, Denmark; andre.briend@gmail.com; 9Center for Child Health Research, Faculty of Medicine and Medical Technology, Tampere University, 33100 Tampere, Finland

**Keywords:** severe acute malnutrition (SAM), community health workers (CHW), integrated community case management (iCCM), mid-upper arm circumference (MUAC), coverage

## Abstract

Geographical and economic access barriers to health facilities (HF) have been identified as some of the most important causes of the low coverage of severe acute malnutrition (SAM) treatment. The objective of this study is to assess the effectiveness and coverage of SAM treatment delivered by community health workers (CHWs) in the Guidimakha region in Mauritania, compared to the HF based approach. This study was a nonrandomized controlled trial, including two rural areas. The control group received outpatient treatment for uncomplicated SAM from HF, whilst the intervention group received outpatient treatment for uncomplicated SAM from HF or CHWs. A total of 869 children aged 6–59 months with SAM without medical complications were included in the study. The proportion of cured children was 82.3% in the control group, and 76.4% in the intervention group, we found no significant difference between the groups. Coverage in the intervention zone increased from 53.6% to 71.7%. In contrast, coverage remained at approximately 44% in the control zone from baseline to end-line. This study is the first to demonstrate in Mauritania that the decentralization model of CHWs treating SAM improves acute malnutrition treatment coverage and complies with the international quality standards for community treatment of acute malnutrition. The non-randomized study design may limit the quality of the evidence, but these results could be used by political decision-makers as a first step in revising the protocol for acute malnutrition management.

## 1. Introduction

Food insecurity and malnutrition are often high and widespread, with seasonal peaks pushing millions into crisis in Mauritania [1]. The 2019 SMART survey indicated that the national nutrition situation remains serious, with a prevalence of global acute malnutrition (GAM), defined by a weight-for-height z-score (WHZ) of less than −2 or the presence of oedema of 11.2%, and a prevalence of severe acute malnutrition (SAM), defined as a WHZ of less than −3 or oedema, of 1.8% [2]. Children suffering from SAM have an increased risk of serious illness and death, primarily from acute infectious diseases [3,4,5,6]. In the Guidimakha region, located in the extreme south of Mauritania, the GAM prevalence in 2019 was 17%, and SAM was 2.9%, which remains a serious situation and one of the worst in the country [2]. This prevalence was higher than the emergency thresholds defined by the World Health Organization by a prevalence of GAM superior to 10% [7].

Geographical and economic access barriers to health facilities (HF) have been identified as one of the most important causes of the low coverage of malnutrition treatment [8]. The geographical distribution of health structures covering a radius of 0–5 km, is unequal in Mauritania. This deprives populations of the contribution that health workers could make in the fight against malnutrition [9]. Action Against Hunger carried out a survey in 2019 to evaluate the coverage of SAM treatment in Guidimakha. Distance to the HF was shown to be one of the main reasons why people do not have access to SAM treatment. This situation has become evident in the Nutrition Causal Analysis (NCA) performed in the area [10]. The isolation of certain localities in Guidimakha makes it challenging to access essential services. Among the recommendations were suggestions to improve geographical access to health posts and health centres and increase health-related activities in the most remote villages [10].

In 2017, Mauritania identified the community approach through the training of community health workers (CHWs) and the reactivation of community health units as one of the major actions to ensure that all people have access to health services. These CHWs, according to government policy, are responsible for the promotion of good hygiene and feeding practices, the treatment of malaria, diarrhoea, and pneumonia; and to identify and refer children suffering acute malnutrition in the community using mid-upper arm circumference (MUAC) [11].

At the international level, during the COVID-19 pandemic, UNICEF and the Global Nutrition Cluster have made a strong recommendation that countries should adopt different modalities of simplified approaches to increase coverage of acute malnutrition and reduce the pandemic’s impact on malnourished children. Among these actions, the decentralization of SAM treatment with CHWs has been included [12]. Evidence regarding the effectiveness of this approach is emerging [13]. However, most of these results were related to small pilot studies. Further analysis is needed to evaluate the quality of care and coverage in other contexts and how to adapt and integrate the approach into the country’s policies.

This study tested the hypothesis that the integration of SAM treatment as part of the iCCM package currently delivered by CHWs in Mauritania will improve the early identification of cases and better access to treatment services, and clinical outcomes of SAM treatment (including cure, death and in particular defaulter) will improve. The objective of this study is to assess the effectiveness and coverage of SAM treatment delivered by CHWs in the Guidimakha region in Mauritania compared to the HF based approach.

## 2. Materials and Methods

This study was a nonrandomized controlled trial conducted in an agro-pastoral region of Guidimakha. The control zone involved the communes of Ould Yengé and Dafor in the Department of Ould Yengé and the commune of Baydiam of the Khabou Department with a total population of 35,562 habitants. The intervention zone included the communes of Sélibabi and Hassi Cheggar in the Department of Sélibabi with a total population of 44,885 habitants [14]. Children in the control zone received outpatient treatment for uncomplicated SAM from HF, while in the intervention group zone they received outpatient treatment for uncomplicated SAM from HF or CHWs. The intervention area had 10 HF and 12 CHWs, and the control area included 6 HF. To assess the comparability of both zones, a two-stage cross-sectional cluster survey was conducted before implementing the intervention. The first level of sampling consisted of villages covered by HF in each zone, and the second level consisted of households in selected villages. Data collection took place from 7 June to 25 June 2018. NutriSurvey.ena delta software was used to calculate the needed sample size based on the prevalence of SAM as measured by MUAC, which was estimated to be 4.9%, desired level of accuracy 3%, design effect of 1.5 with an average household size of 5.5, 16% of children under five years and 6% of non-response household [15,16]. The total number of households to be surveyed was 218 in each zone (30 clusters with seven households each). When a household was selected, all children aged 6 to 59 months were included for MUAC measurements and oedema testing and demographic and socio-economic variables described in Table 1 were collected.

The CHWs in the intervention zone received 21 days of training based on the basic health assistance package of integrated community case management (iCCM), using the training module of the Ministry of Health. The trainers were the health district management team together with Action Against Hunger staff. This training included health promotion, infant and young child feeding (IYCF) practices, hygiene practices, family planning, neonatal care, and management of diarrhoea, malaria, and pneumonia. As treatment of acute malnutrition is not considered part of the activities that CHWs in Mauritania must carry out, Action Against Hunger supported the Ministry to do this part and ensure the quality of care. A pre- and post-test was administered to all participating CHWs to ensure that the knowledge had been acquired correctly. During the study, CHWs received periodic supportive supervision by the healthcare-responsible staff from the HF and Action Against Hunger supervisors. Training modules can be found in Appendix A.

The study took place between November 2018 and July 2019. The inclusion and exclusion criteria defined in the national protocol for the management of acute malnutrition were applied [17]. All children aged 6 to 59 months presented to a HF or to a CHW’s site, and/or detected by community volunteers (Relais Commaunitaires in French) or mobile clinics with mild or moderate oedema (+, ++), a MUAC less than 115 mm, and/or a WHZ less than −3 were included in the analysis. All severe oedema cases (+++), children with other severe medical conditions, or those who failed the appetite test were referred for inpatient treatment [18,19]. Non complicated cases were treated at home. These children received 170 kcal/kg/day of ready to use therapeutic food (RUTF) to be consumed at home. The children were rechecked once a week until they reached one of the program’s exit criteria, MUAC > 125 mm, and/or WHZ > 1.5. They also received amoxicillin (50–100 mg/kg/day divided twice a day for five days) and one single dose of 500 mg of mebendazole at the first visit for deworming.

To compare treatment effectiveness between the two zones, outcomes of treated children were examined. The primary outcome was the proportion of cured children, defined by an absence of oedema and WHZ equal to or greater than 1.5 or/and MUAC > 125 mm during two consecutive weeks. Secondary outcomes were the proportion of defaulters defined the child being absent at two follow-up visits and the proportion of nonresponse defined by proportion of children not reached recovery after three months of treatment. Medical referral was considered when severe signs of illness appeared, oedema did not disappear, absence of weight gain in non-oedematous patients for 21 days or weight loss. Disaggregated information was collected from each child from the patient cards. The WHZ value was calculated through WHO Anthro software with the recorded anthropometric data of weight and height [20]. The length of stay in treatment was calculated from the date of admission and the date of discharge. The number of RUTF sachets received during the entire treatment was also recorded.

To assess the possible impact of treatment by CHWs on the treatment coverage, two surveys were conducted in each zone at the baseline (June 2018) and end-line (June 2019) of the study applying the SQUEAC (semi quantitative evaluation of access and coverage) standardized methodology [21]. The statistical analysis was performed with SPSS v.26. The normality of the continuous variables was tested with the Shapiro–Wilk test. Depending on the result, the central parameters were compared with Student’s t-test for normal and the Mann–Whitney test for non-normal parameters. The comparison of percentages was made through crossed tables, applying the chi-square statistic with Yates’ correction when the expected cases were less than 5 in more than 20% of the cells. The Mantel-Haenszel chi-square test was applied to compare the final treatment coverage adjusted to the baseline coverage data. For the analysis of the treatment outcomes, a Cox regression analysis was performed to obtain the time-adjusted probability (hazard ratio) until the outcome occurred. A 95% confidence level applied in all analyses, considering significant *p* values below 0.05.

A steering committee was set up in Nouakchott, composed of the responsible staff from the Mauritanian Ministry of Health, academics of Nouakchott University, a team from the Public Health Research Institute, NGOs, and United Nations agencies such as WHO and UNICEF. The objectives of this committee were to rigorously monitor the operational implementation of the project and make recommendations, if necessary, discuss the evolution of indicators related to the research, and ensure that the technical aspects of the research protocol were respected. Informed consent was sought from all participants, both from the socioeconomic and coverage surveys and parents or guardians of children included in the study. The study received approval from the Ethical Committee of the Ministry of Health in Nouakchott, 25 October 2018.

## 3. Results

The results of the baseline socioeconomic survey are shown in Table 1. Study zones did not differ in demographic variables, acute malnutrition prevalence, or house characteristics, except for the house floor, in which there was a higher proportion of cement floors in the intervention zone (24.5% [18.9–30.9] vs. 7.6% [4.5–11.9] (*p* < 0.001)), showing better conditions. There were also no differences in the main barriers to access to health. However, they differed in terms of treatment preferences for the sick child: the control group resorted more to traditional healers (25.1% vs. 8.5%, *p* < 0.001), while the intervention zone resorted more to formal health structures (80.2% vs. 55.6%, *p* < 0.001).

At the beginning of the study, prevalence of acute malnutrition in children under five was similar in both zones, and there was no difference between coverage of SAM treatment.

In the intervention group 618 children were treated and in the control group 251. Among treated children, there were no differences in sex proportion (females in intervention: 56.5%, *n* = 350; in control: 52.2%, *n* = 131; *p* = 0.252) or age of children included (intervention: 15.7 ± 8.9 months, control: 15.3 ± 8.3 months, *p* = 0.631), with most children under 24 months (intervention: 90.2%, *n* = 559; control: 93.2%, *n* = 234; *p* = 0.151).

Most children in both groups came from active screening in the community without differences between groups (intervention: 59.0% vs. control: 54.4%, *p* = 0.211). Spontaneous referrals (children who arrived directly to the HF or CHWs to receive malnutrition treatment) were higher in the control group (43.2% vs. 33.2%, *p* = 0.006); and passive screening cases (children who arrived at the HF or CHWs, to receive treatment for other disease and was identified as malnourished) were higher in the intervention group (3.8% vs. 0.4%, *p* = 0.006). No differences were found for patients referred from other structures (control: 2.0% vs. 3.9%, *p* = 0.151) or for active screening where most of the cases treated came from (control: 54.4% vs. 59.0%, *p* = 0.211).

No significant differences were found in the proportion of cases with oedema between the control (4 cases) and intervention groups (1 case) (0.4% vs. 0.7%, *p* = 0.956). The CHWs did not register any cases of oedema within the intervention group. Table 2 shows the results regarding anthropometric severity at admission. No significant differences were found between the control and intervention groups for the median values of MUAC or WHZ or the proportion of children in the most severe ranges of both anthropometric indicators. This section may be divided by subheadings. It should provide a concise and precise description of the experimental results, their interpretation, as well as the experimental conclusions that can be drawn.

However, inside the intervention group, children who attended HF to be treated by nurses, showed a significantly lower MUAC than those treated by the CHWs. Additionally, the number of cases with a MUAC lower than 110 mm was five times lower among CHWs (1.9% vs. 10.5%) than in HF (Table 3). The remaining variables showed no significant differences between groups.

Treatment outcomes are summarized in Table 4. Among children seen on admission, treatment outcome was available to 81.1% (705/869). Outcomes were unknown for 42 children (16.7% of total admission) in the control group, and for 122 (19.7%) in the intervention group. No significant differences were found between the control and intervention groups for any of the treatment exit reasons. The proportion of cured, defaulters and deaths in the intervention group met the established international standards of >75%, <15%, and <10% [22]. There were no deaths or children who did not respond to nutritional treatment, indicating good quality of performance. There were no differences in the cured children’s anthropometric evolution, with an average weight gain of 4.7 g/kg/day.

Within the intervention group, the treatment results obtained by the CHWs were analysed comparatively with the health staff in the HF group (Table 5). There were also no significant differences found in the proportion of cured children or defaulters. The proportion of cases that needed to be referred for medical complications was lower with CHWs, although this difference did not reach statistical significance. A higher proportion of internal transfers, to other HF and/or CHWs site to follow up and finish their treatment, was found in the children treated by CHWs, the difference was statistically significant. There was a more significant gain in total MUAC in patients treated in HF, but that difference disappeared when calculating the average daily gain.

Among children discharged as cured, the recovery time was 45 days. No significant differences were found between the control and intervention groups (45.3 vs. 45.8, *p* = 0.969). Within the intervention group, children treated by the CHWs, recovered more quickly than those children treated at HFs (37.3 vs. 46.2, *p* = 0.004). The proportion of children in treatment longer than six weeks within the intervention group, was lower in children treated by CHWs comparing with those treated at HF (28.3% vs. 50.9%, *p* = 0.001). In line with the treatment time, the total consumption of RUTF was significantly lower in children treated by CHWs in the intervention group (93.4 vs. 110.3, *p* = 0.035).

The results of the treatment coverage surveys are shown in Figure 1. At the beginning of the study, the coverage did not differ significantly between zones. After a year of incorporating CHWs as treatment providers at the community level outside the HF, coverage in the intervention zone increased significantly from 53.6% to 71.7%. In contrast, coverage remained at approximately 44% in the control zone from baseline to end-line which does not reach the 50% established internationally as the minimum standard for treatment coverage in rural areas [22]. This final coverage between the control and intervention groups is statistically significant after adjusting for initial coverage (Mantel-Haenszel Chi-square *p* = 0.012). 

## 4. Discussion

These findings regarding treatment outcomes are consistent with the evidence emerging from several small-scale pilot studies regarding the effectiveness and coverage of CHWs treating SAM. Wilunda et al. in a non-inferior-quasi experimental study in Tanzania, comparing treatment outcomes between CHWs and HFs, showed that CHWs obtained a proportion of cured children of 90.5% and a proportion of defaulters of 6.5% [23] Kozouki et al. working with low literacy CHWs in South Soudan, reached 75% of children cured of SAM [24]. In this present study, we found no significant differences in terms of cured and defaulted proportion in children between the control and intervention group. The results in both groups comply with the international Sphere standard that establishes that the cured proportion should be above 75% and defaulter proportion under 15% of the total of discharged children [22] López Ejeda et al., in a review of operational experience about CHWs, analysed 12 peer-review and eight grey literature articles, where CHWs reached cured and defaulter proportions of children similar to those mentioned above [25].

Data on the effect of treatment of SAM by CHW in coverage were less often reported; only three of these studies included coverage assessments, those located in Bangladesh [26], Angola [27], and Mali [28]. In Bangladesh, the CHW-based intervention achieved 89% coverage, one of the highest coverage rates recorded by an SAM treatment intervention in the scientific literature. In Mali, after one year of project implementation, the intervention group reached 86.7% coverage. In Angola, the coverage was estimated at 82.1% in the areas where CHWs treated acute malnutrition. These results are similar to the present study, where we obtained 71.7% coverage in the intervention group, a massive increase compared with the control group, which observed no significant changes of treatment coverage after one year.

In this study, there was a large difference in number of children treated in the two groups: 618 in the intervention group vs. 251 in the control group, a 146% difference. Total population in the intervention group was 44,885 habitants, and 35,562 habitants in the control group, 26% higher in the intervention group. A difference in incidence between the two zones cannot be excluded, but we have no direct measure of SAM incidence in the two zones, and we have no reason to believe this was the explanation. Pre and post survey in the intervention group suggested a large increase in coverage from 44.2 to 71.7%, a 62% increase. Both pre and post intervention coverages were estimated with large confidence intervals, and a larger increase than the point estimate is plausible. The increase coverage may be the main explanation of the larger number of treated children in the intervention area. This result is consistent with our hypotheses that CHWs can increase coverage of acute malnutrition treatment. Zulu and Perry 2021, in a recent series found that the large-scale involvement of CHWs in the health care system has a huge potential for accelerating universal health coverage and improving population health [29].

We found no significant difference in median MUAC and WHZ between the control and intervention groups on admission. However, when we analysed the CHWs and the HFs separately within the intervention group, there was a significant difference between the median MUAC of 116 mm and 115 mm in the HF group. The percentage of children in the lowest quartile of MUAC (<110 mm) was also lower within the CHWs, 1.9% vs. 10.5% in HF on admission. These results are consistent with the hypothesis of an earlier case detection resulting from the increased coverage of screening. Our results with MUAC are consistent with those found by Lopez-Ejeda et al. in Mali, where children treated by CHWs in the lowest quartile were 18% vs. 32.4% at HF, and the median MUAC at admission was 115 mm compared to 114 mm at HF [30]. No difference was found in our study when considering the WHZ criteria at admission.

The work presented here has some limitations. This study was not a randomized controlled trial (RCTs), and thus, we could not rule out that our results were due, to some extent, to differences among areas independent of our intervention. The follow-up period was short and confounding factors could not be fully regulated. The study area was supported by an international NGO and hence, may reflect performance levels associated with well-supported interventions.

Randomized controlled trials (RCTs) are rightly regarded as the gold standard for clinical decision-making purposes, especially in evaluating new drugs or dosages. However, randomization is often not feasible in the evaluation of public health interventions in low-resource settings. Randomization, without further analyses for adequacy and plausibility, is not sufficient to support public health decision making, regardless of the level of the power of the statistical significance achieved. An intervention that works well in a given setting proved by an RCT may be ineffective elsewhere, presenting a huge challenge to international health recommendations. Proper evidence-based public health intervention must rely on a variety of types of evidence, often in combination [31,32].

## 5. Conclusions

This study is the first to demonstrate in Mauritania that the decentralization model of CHWs treating SAM improves acute malnutrition treatment coverage and complies with the international quality standards for community treatment of acute malnutrition. Coverage is key in determining the impact of SAM treatment [33,34]. This evidence could be used by political decision-makers as a first step in revising the protocol for acute malnutrition management within primary health care policy. These results are in line with the evidence found in other countries indicating that the integration of SAM treatment into the package of activities of CHWs can effectively contribute to universal health coverage.

Our results open the door to new studies applying other simplified approach protocols [35]. Combining these protocols with decentralized treatment with CHWs would have great potential in emergency contexts such as the current COVID-19 pandemic and isolated or insecure areas with limited access to health services.

## Figures and Tables

**Figure 1 children-08-01132-f001:**
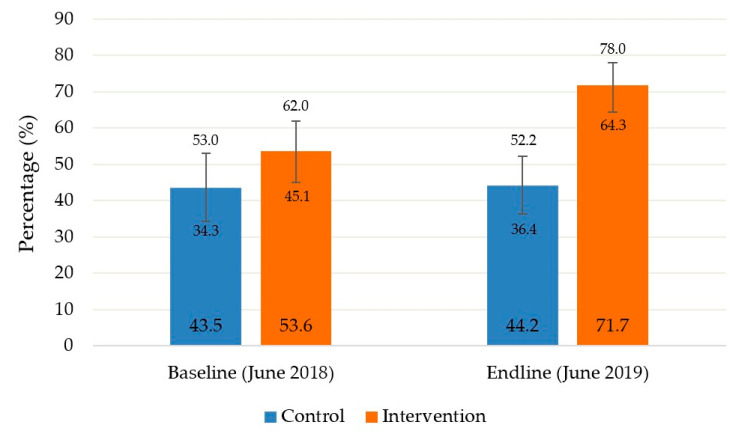
Baseline and end-line coverage assessment of noncomplicated severe acute malnutrition treatment compared by study groups.

**Table 1 children-08-01132-t001:** Socioeconomic characteristics of the population in study zones.

	Control		Intervention		*p* Value
	*n*	% (95% C.I.)	*n*	% (95% C.I.)	
**Demographics**	724		730		
Female proportion	310	42.8 (40.2–45.5)	332	45.5 (42.8–48.1)	0.307
6–59 month children	501	69.2 (65.7–72.6)	518	71.0 (67.7–74.3)	0.464
Global Acute Malnutrition	36	7.2 (5.2–9.8)	34	6.6 (4.7–9.0)	0.695
Severe Acute Malnutrition	2	0.4 (0.1–1.4)	2	0.4 (0.4–1.4)	0.973
**House characteristics**	223		212		
Cement floor	17	7.6 (4.5–11.9)	52	24.5 (18.9–30.9)	<0.001
Handmade earth brick roof	50	22.4 (17.1–28.5)	36	17.0 (12.2–22.7)	0.154
House in property	197	88.3 (83.4–92.2)	179	84.4 (78.8–89.0)	0.234
Potable water in the house	12	5.4 (2.8–9.2)	18	8.5 (5.1–13.1)	0.201
Potable water closes to household	111	49.8 (43.0–56.5)	92	43.4 (36.6–50.4)	0.185
**Health care access barriers**	223		212		
Cost	112	50.2 (0.43–0.57)	101	47.6 (40.8–54.6)	0.590
Distance	107	48.0 (41.3–54.8)	105	49.5 (42.6–56.5)	0.747
Family disagrees	4	1.8 (0.5–4.5)	6	2.9 (1.1–6.1)	
**Sick child treatment preference**	223		212		
Medication of health center	124	55.6 (48.8–62.2)	170	80.2 (74.2–85.3)	<0.001
Traditional self-medication (herbs)	15	6.8 (3.8–10.9)	9	4.2 (2.0–7.9)	0.257
Self-medication (street drugs)	28	12.5 (8.5–17.6)	15	7.1 (4.0–11.4)	0.556
Traditional medicine	56	25.1 (19.6–31.3)	18	8.5 (5.1–13.1)	<0.001

**Table 2 children-08-01132-t002:** Anthropometric measurements at admission by study group.

Study Groups	Control	Intervention	*p* Value
**MUAC indicators**	*n* = 251 median (IQR)	*n* = 618 median (IQR)	
MUAC (mm)	112 (115–120)	112 (115–120)	0.478
MUAC quartiles *	% (*n*)	% (*n*)	
Q1 < 110 mm	8.4 (21)	9.1 (56)	0.744
Q2 ≥ 110 mm to <115 mm	32.3 (81)	31.7 (196)	0.873
Q3 ≥ 115 mm to <120 mm	24.7 (62)	26.4 (163)	0.610
Q4 ≥ 120 mm	34.7 (87)	32.8 (203)	0.030
**Weight-for-Height indicators**	*n* = 239 median (IQR)	*n* = 601 median (IQR)	*p* value
Weight (kg)	6.70 (5.90–7.50)	6.70 (5.95–7.50)	0.854
Height (cm)	72.0 (67.0–77.0)	72.0 (67.5–77.0)	0.374
WHZ	−3.26 (−3.61–−2.80)	−3.31 (−3.84–−2.78)	0.135
WHZ ranges *	% (*n*)	% (*n*)	
Q1 < −3.76	20.9 (50)	27.3 (164)	0.056
Q2 ≥ −3.76 to <−3.29	25.5 (61)	24.5 (147)	0.747
Q3 ≥ −3.29 to <−2.78	29.3 (70)	24.1 (145)	0.122
Q4 ≥ −2.78	24.3 (58)	24.1 (145)	0.966

* Quartile values calculated for the whole sample (control + intervention). IQR: Interquartile range; MUAC: Middle-Upper Arm Circumference; WHZ: Weight for Height z-score. WHZ measure was unknown for 12 children in the intervention group and 17 children in the control group.

**Table 3 children-08-01132-t003:** Anthropometric measurements at admission by treatment provider within the intervention group.

Intervention Group	Health Staff	CHWs	*p* Value
**MUAC indicators**	*n* = 512 median (IQR)	*n* = 106 median (IQR)	
MUAC (mm)	115 (111–120)	116 (114–120)	0.015
MUAC quartiles *	% (*n*)	% (*n*)	
Q1 < 110 mm	10.6 (54)	1.9 (2)	0.005
Q2 ≥ 110 mm to <115 mm	32.0 (164)	30.2 (32)	0.711
Q3 ≥ 115 mm to <120 mm	25.4 (130)	31.1 (33)	0.222
Q4 ≥ 120 mm	32.0 (164)	36.8 (39)	0.623
**Weight-for-Height indicators**	*n* = 496 median (IQR)	*n* = 105 median (IQR)	*p* value
Weight (kg)	6.70 (5.90–7.48)	6.80 (6.15–7.80)	0.179
Height (cm)	72.0 (67.0–77.0)	75.2 (98.7–78.0)	0.370
WHZ	−3.31 (−3.86–−2.79)	−3.31 (−3.75–−2.63)	0.366
WHZ quartiles *	% (*n*)	% (*n*)	
Q1 < −3.76	27.8 (138)	24.8 (26)	0.522
Q2 ≥ −3.76 to <−3.29	24.0 (119)	26.7 (28)	0.565
Q3 ≥ −3.29 to <−2.78	25.4 (126)	18.1 (19)	0.112
Q4 ≥ −2.78	22.8 (113)	30.5 (32)	0.094

* Quartile values calculated for the whole sample. IQR: Interquartile range; MUAC: Middle-Upper Arm Circumference; WHZ: Weight for Height z-score.

**Table 4 children-08-01132-t004:** Treatment outcomes and anthropometric improvement of children compared by study groups.

Whole Sample	Control (*n* = 209)		Intervention (*n* = 496)		Comparison
**Treatment outcomes**	** *n* **	**%**	** *n* **	**%**	**HR (95% C.I.); *p* value ^2^**
Cured	172	82.3	379	76.4	0.967 (0.807–1.159); 0.719
Default	8	3.8	18	3.6	0.915 (0.395–2.121); 0.836
Nonrespondent	0	0	0	0	
Medical reference	20	9.6	67	13.5	1.297 (0.733–2.294); 0.732
Internal transference	9	4.3	32	6.5	1.659 (0.760–3.625); 0.204
Death	0	0	0	0	
**Anthropometric gain ^1^**	** *n* **	**Median (IQR)**	** *n* **	**Median (IQR)**	***p* value ^3^**
Total weight (g/kg)	161	197.2 (157.9–254.3)	356	209.7 (164.6–255.2)	0.283
Daily weight (g/kg/day)	161	4.68 (3.17–7.11)	356	4.73 (3.39–7.35)	0.426
Total MUAC (mm)	168	11.0 (8.0–15.0)	364	13.0 (9.0–16.0)	0.059
Daily MUAC (mm/day)	168	0.27 (0.17–0.41)	363	0.29 (0.20–0.43)	0.139

^1^ Considering only those children discharged as cured and excluding oedema cases; ^2^ Crude coefficients; ^3^ Mann–Whitney Test; C.I.: Confidence Interval; HR: Hazard Ratio; IQR: Interquartile Range; MUAC: Middle-Upper Arm Circumference. Treatment outcome was unknown for 122 children in the intervention group and 42 children in the control group.

**Table 5 children-08-01132-t005:** Treatment outcomes and anthropometric improvement of children within the intervention group compared by the treatment provider.

Intervention Group	HEALTH STAFF (*n* = 418)		CHWs (*n* = 78)		Comparison
**Treatment outcomes**	** *n* **	**%**	** *n* **	**%**	**HR (95% C.I.); *p* value ^2^**
Cured	319	76.3	60	76.9	1.135 (0.860–1.498); 0.373
Default	16	3.8	2	2.6	0.384 (0.051–2.909); 0.354
Nonrespondent	0	0	0	0	
Medical reference	64	15.3	3	3.8	0.246 (0.060–1.019); 0.053
Internal transference	19	4.5	13	16.7	4.436 (2.146–9.170); <0.001
Death	0	0	0	0	
**Anthropometric gain ^1^**	** *n* **	**Median (IQR)**	** *n* **	**Median (IQR)**	***p* value ^3^**
Total weight (g/kg)	298	211.4 (163.9–261.5)	58	196.2 (168.4–232.5)	0.403
Daily weight (g/kg/day)	298	4.63 (3.35–7.54)	58	5.49 (3.76–8.38)	0.056
Total MUAC (mm)	305	13.0 (9.5–17.0)	59	12.0 (7.0–14.0)	0.011
Daily MUAC (mm/day)	305	0.29 (0.19–0.43)	59	0.29 (0.21–0.48)	0.749

^1^ Considering only those children discharged as cured and excluding oedema cases; ^2^ Crude coefficients; ^3^ Mann–Whitney Test; C.I.: Confidence Interval; HR: Hazard Ratio; IQR: Interquartile Range; MUAC: Middle-Upper Arm Circumference. Treatment outcome was unknown for 28 children treated by CHWs and 94 children treated at the HF level.

## Data Availability

The data presented in this study are available on request from the corresponding author.

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
