# Peer review of "Effectiveness and Coverage of Treatment for Severe Acute Malnutrition Delivered by Community Health Workers in the Guidimakha Region, Mauritania"

_children, 2021, doi:10.3390/children8121132_

Round 1

Reviewer 1 Report

This is my review on the study entitled “Effectiveness and coverage of treatment for severe acute malnutrition delivered by community health workers in the Guidimakha region, Mauritania”. The purpose of the study is clearly stated and I believe is rather significant as severe acute malnutrition should not be covered only by the “basic” health facilities. Introduction is sufficient justifying the purpose of the investigation. Material and methods are thoroughly described. Results are clearly presented, indicating the effectiveness of community health workers’ regarding the increasing coverage of SAM. I agree with the limitation mentioned on discussion but all in all this is an interesting study that could be published. I would suggest to remove the citation and “small” discussion from the conclusions to the discussion section and structure a concise conclusion of few lines.

Author Response

Dear reviewer,

We sincerely thank you for taking the time to do this review and for the questions submitted which allows us to improve our work for a better understanding of future readers.

Your suggestions have been considered and the request adjustment has been made in the paper with control changes. In the attached document, we have explained the answers to improve comprehension, with the lines where we have included any change.

Thank you again for your time and dedication.

Best regards,

This is my review on the study entitled “Effectiveness and coverage of treatment for severe acute malnutrition delivered by community health workers in the Guidimakha region, Mauritania”. The purpose of the study is clearly stated and I believe is rather significant as severe acute malnutrition should not be covered only by the “basic” health facilities. Introduction is sufficient justifying the purpose of the investigation. Material and methods are thoroughly described. Results are clearly presented, indicating the effectiveness of community health workers’ regarding the increasing coverage of SAM. I agree with the limitation mentioned on discussion but all in all this is an interesting study that could be published. I would suggest to remove the citation and “small” discussion from the conclusions to the discussion section and structure a concise conclusion of few lines.

Authors: Thank you very much for your review and comments. We have made the request adjustment. It has been removed the paragraph from conclusion to the discussion section lines 299-301. This make smaller the conclusion, just with two main focus in: CHWs can increase coverage of acute malnutrition and outcomes indicators reach the international sphere standards; second, the study results open the door to further research of CHWs initiative together with other simplified approaches.

Reviewer 2 Report

Comments are provided as follows:

  1. This was unfortunately a non-randomized study (in a non-cluster randomized or non-individually randomized manner), which could suffer from biases and confounding influences. Thus, this was a pilot study and the application of the results to political evidence should be modestly stated.
  2. The primary outcome of cured population was not different between intervention and control groups. This should be described in Abstract (as hazard ratio) and discussed more in the Discussion part.
  3. Throughout the text, the contents of severe acute malnutrition, SAM, and acute malnutrition appeared mixed. The prior findings and knowledge of SAM and simply acute malnutrition could differ in several aspects. Th data on SAM could not always be expanded to acute malnutrition. The authors can review the overall text and the text may be shorter in focusing on SAM.
  4. In the last paragraph of Introduction, the author can state the working hypothesis for the study.
  5. In Methods, the authors must detail the policy and the validity of the area’s selection for the study.
  6. In Methods, the authors might cite the proper references on the validity of training for workers. This was important because it affected the results of intervention. This was also important when the protocol can be made by the other investigators.
  7. In Table 1, what was ‘healer’?
  8. In Table 2, the space was required before ‘mm’ (e.g., change 110mm to 110 mm).
  9. In Tables, the expressions […] and (…) were mixed.
  10. As the study limitations, the follow-up period was short and moreover, confounding factors were not fully regulated.

Author Response

Dear Reviewer,

We sincerely thank you for taking the time to do this review and for the questions submitted which allows us to improve our work for a better understanding of future readers.

Your suggestions have been considered and the request adjustment has been made in the paper with control changes. In the reviewers' sheet, we have explained the answers to improve comprehension, with the lines where we have included all the changes.

Thank you again for your time and dedication.

Best regards,

  1. This was unfortunately a non-randomized study (in a non-cluster randomized or non-individually randomized manner), which could suffer from biases and confounding influences. Thus, this was a pilot study and the application of the results to political evidence should be modestly stated.

Authors: Thank you very much for your comment. You are right this is not an RCT, and results must be taken with caution, we are aware of the study limitations and this is what we have tried to explain in the paper.

In the abstract it has been used “… is the first step”, not a categorical statement. In discussion line 312, 313   “… we could not rule out that our results were due to some extent to differences among areas independent of our intervention”

In conclusions, it is mentioned line 320, 321 “This evidence could be used by political decision-makers as a first step in revising the protocol for acute malnutrition management within primary health care policy” again, not categorical.

Line 330, 331 is mentioned “….Our results open the door to new studies…”, which means further analyses could be need , to support decision making of the Ministry of Health.

  1. The primary outcome of cured population was not different between intervention and control groups. This should be described in Abstract (as hazard ratio) and discussed more in the Discussion part.

Authors: Abstract: The request adjustment has been made. Added, We found no significant difference was found

Line 270-276 this paragraph has been added. In the present study we have found not significant difference in terms of cured and defaulted proportion in children between the control and intervention group. The results in both groups comply with the international Sphere standard that establishes that the cured proportion should be above 75% and defaulter proportion under 15% of the total of discharged children

  1. Throughout the text, the contents of severe acute malnutrition, SAM, and acute malnutrition appeared mixed. The prior findings and knowledge of SAM and simply acute malnutrition could differ in several aspects. Th data on SAM could not always be expanded to acute malnutrition. The authors can review the overall text and the text may be shorter in focusing on SAM.

Authors: Thank you very much for your suggestion. Modifications has been done in the whole paper.

  1. In the last paragraph of Introduction, the author can state the working hypothesis for the study.

Authors: The hypothesis of the present research has been added. Line 82-85. We have hypothesis that the integration of SAM treatment as part of the iCCM package currently delivered by CHWs in Mauritania will provide improve in early identification of cases and better access to treatment services, and clinical outcomes of SAM treatment (including cure, death and in particular defaulter) will improve.

  1. In Methods, the authors must detail the policy and the validity of the area’s selection for the study.

Authors: As this was not an RCT, at the beginning of the study, two health areas within the same region that had a similar number of SAM admissions in the preceding 5 years. Before starting the admission of SAM children, a socio-economic survey was conducted in a selected number of households (the methodology explained in the paper), where at the same time anthropometric measurements of children under 5 years of age were taken to estimate the prevalence of SAM at the time of the study. Line 99-109. Table 1, explain all the variables take into consideration to compare both studies areas.

  1. In Methods, the authors might cite the proper references on the validity of training for workers. This was important because it affected the results of intervention. This was also important when the protocol can be made by the other investigators.

Authors: Thank you very much to address this part. Training modules used in the country has been included as supplementary materiel.

Line 122. Training modules can be found in supplementary materials.

  1. In Table 1, what was ‘healer’?

Authors: Healers are the human resources responsible for traditional medicine. It has been changed the name for a better understanding.

  1. In Table 2, the space was required before ‘mm’ (e.g., change 110mm to 110 mm).

Authors: We have made the request adjustment.

  1. In Tables, the expressions […] and (…) were mixed.

Authors: We have made the request adjustment in table 2, 4 and 5.

  1. As the study limitations, the follow-up period was short and moreover, confounding factors were not fully regulated.

Authors: Thank you very much for this, there was a mistake that has been modified in line 123, even like this, the period was shorter than 1 year. The request has been made. Line 315-316

Round 2

Reviewer 2 Report

The paper was improved.

  1. In Abstract, the authors can describe that this was not a randomized study. The conclusion may be more modestly stated as the conclusion in main text.
  2. In Methods, the authors can add the justification of non-randomized design for making evidence-based policy with some references.

Author Response

Dear reviewer,

Thank you very much again, for your time and dedication with this second review to improve the quality and understanding of our paper.

Reviewer: In Abstract, the authors can describe that this was not a randomized study. The conclusion may be more modestly stated as the conclusion in main text.

Authors: The request adjustment have been made “The non-randomized study design may limit the quality of the evidence, but these results could be used by political decision-makers as a first step in revising the protocol for acute malnutrition management”

Reviewer: In Methods, the authors can add the justification of non-randomized design for making evidence-based policy with some references.

Authors: We have included some references which can support when and why non RCT can be made. As the references has not been used as part of our methodology design, we have considerer to include all the paragraph in limitations of the study. This request adjustment have been made. “Randomized controlled trials (RCTs) are rightly regarded as the gold standard for clinical decision-making purposes, especially evaluating new drugs or dosages. However, randomization is often not feasible in the evaluation of public health interventions in low-resource settings. Randomization, without further analyses for adequacy and plausibility, is not sufficient to support public health decision making, regardless of the level of the power of the statistical significance achieved. An intervention that works well in a given setting proved by an RCT may be ineffective elsewhere, presenting a huge challenge to international health recommendations. Proper evidence-based public health intervention must rely on a variety of types of evidence, often in combination”  

Thank you again for your dedication

Kind regards